# Physical Fitness and Self-Rated Health in Children and Adolescents: Cross-Sectional and Longitudinal Study

**DOI:** 10.3390/ijerph17072413

**Published:** 2020-04-02

**Authors:** Carmen Padilla-Moledo, Jorge DR Fernández-Santos, Rocio Izquierdo-Gómez, Irene Esteban-Cornejo, Paula Rio-Cozar, Ana Carbonell-Baeza, Jose Castro-Piñero

**Affiliations:** 1GALENO research group, Department of Physical Education, Faculty of Education Sciences, University of Cadiz, 115010 Puerto Real, Spainrocio.izquierdo@uca.es (R.I.-G.); pdelrioc@gmail.com (P.R.-C.); ana.carbonell@uca.es (A.C.-B.); jose.castro@uca.es (J.C.-P.); 2Biomedical Research and Innovation Institute of Cádiz (INiBICA) Research Unit, 11008 Cádiz, Spain; 3PROFITH “PROmoting FITness and Health through physical activity” research group, Department of Physical Education and Sports, Faculty of Sport Sciences, University of Granada, 18071 Granada, Spain; ireneesteban@ugr.es; 4Center for Cognitive and Brain Health, Department of Psychology, Northeastern University, Boston, MA 02115, USA

**Keywords:** cardiorespiratory fitness, muscular strength, motor fitness, physical fitness, self-rated health, children and adolescents

## Abstract

Self-rated health (SRH) is an independent determinant for all-cause mortality. We aimed to examine the independent and combined associations of components of physical fitness with SRH at baseline (cross-sectional) and two years later (longitudinal) in children and adolescents. Spanish youth *(N* = 1378) aged 8 to 17.9 years participated at baseline. The dropout rate at 2-year follow-up was 19.5% (*n* = 270). Participants were categorized as either children (8 to 11.9 years age) or adolescents (12 to 17.9 years age). The ALPHA health- related fitness test battery for youth was used to assess physical fitness, and SRH was measured by a single-item question. Cumulative link, ANOVA and ANCOVA models were fitted to analyze the data. Cardiorespiratory fitness, relative upper body isometric muscular strength, muscular strength score, and global physical fitness were positively associated with SRH in children (OR, 1.048; 95% CI, 1.020–1.076; OR, 18.921; 95% CI, 3.47–104.355; OR, 1.213; 95% CI, 1.117–1.319, and OR, 1.170; 95% CI, 1.081–1.266, respectively; all *p* < 0.001) and adolescents (OR, 1.057; 95% CI, 1.037–1.076; OR, 5.707; 95% CI, 1.122–29.205; OR, 1.169; 95% CI, 1.070–1.278, and OR, 1.154 95% CI, 1.100–1.210, respectively; all *p* < 0.001); and motor fitness was positively associated with SRH only in adolescents at baseline (OR, 1.192; 95% CI, 1.066–1.309; *p* < 0.01). Cardiorespiratory fitness and global physical fitness were positively associated with SRH in children two years later (OR, 1.056; 95% CI, 1.023–1.091; *p* < 0.001; and OR, 1.082; 95% CI, 1.031–1.136; *p* < 0.01; respectively). Only cardiorespiratory fitness was independently associated with SRH in children and adolescents at baseline (OR, 1.059; 95% CI, 1.029–1.090; and OR, 1.073; 95% CI, 1.050–1.097, respectively; both *p* < 0.001) and two years later (OR, 1.075; 95% CI, 1.040–1.112; *p* < 0.001; and OR, 1.043; 95% CI, 1.014–1.074; *p* < 0.01, respectively). A high level of cardiorespiratory fitness at baseline or maintaining high levels of cardiorespiratory fitness from the baseline to 2-year follow-up were associated with a higher level of SRH at 2-year follow-up in children (*p* < 0.01) and adolescents (*p* < 0.05). These findings emphasize the importance of cardiorespiratory fitness as strong predictor of present and future SRH in youth. Intervention programs to enhance cardiorespiratory fitness level of the youth population are urgently needed for present and future youth’s health.

## 1. Introduction

Self-rated health (SRH) is a subjective measure that captures a person’s perception of their overall health status. SRH incorporates psychological, biological, and social dimensions that are unavailable to the external observer, and also provides a dynamic assessment of current health status while integrating a trajectory of personal health [1]. In addition, SRH is a valid proxy for assessing population health across the lifespan. In adults, it is one of the most widely examined measures of health status and has been shown to be a strong predictor of disease and mortality. [2] In youth, it appears to be related to their overall sense of functioning [3] and health-related quality of life [4], however, it deserves more research attention as a health indicator in this population [5].

On the other hand, physical fitness is considered an important health marker, both in the early years and later in life [6,7,8,9]. The main components of physical fitness are cardiorespiratory fitness, muscular strength, and motor fitness [6]. Numerous benefits of components of physical fitness for physical and mental health are well known in youth [6]. Several cross-sectional studies observed that cardiorespiratory fitness was positively associated with SRH in youth [10,11,12,13,14]. Only two studies have examined the association of muscular strength with SRH in youth showing a positive association, [13,15] and there are no studies investigating the association between motor fitness and SRH in youth. Notably, these three components of physical fitness (cardiorespiratory fitness, muscular strength, and motor fitness) are highly associated with one another [16], and it would be necessary to differentiate which components are important in relation to SRH.

Particularly, SRH is an independent determinant for all-cause mortality [17] and a strong predictor of future illness, independently of clinical health status. [18] Therefore, it would be relevant to differentiate which components of physical fitness might be more strongly associated (independent or combined) with SRH later in life in order to use them as a predictor of future SRH level. Only two longitudinal studies in older people have observed a positive association between self-reported global physical fitness [19] and muscular strength [20] with SRH.

To the best of our knowledge, there are no studies investigating the independent cross-sectional association of cardiorespiratory fitness, muscular strength, and motor fitness with SRH in youth, or longitudinal studies investigating the independent or combined associations of cardiorespiratory fitness, muscular strength, and motor fitness with SRH in youth.

The UP&DOWN study is a 3-year longitudinal study conducted in a Spanish sample of children and adolescents [21]. This study was designed to assess the impact of physical activity and sedentary behaviors over time on health indicators. Data from the UP&DOWN study provide an excellent opportunity to study the cross-sectional and longitudinal associations between components of physical fitness and SRH in children and adolescents. Therefore, the purpose of the present study is to examine the independent and combined associations of three components of physical fitness (cardiorespiratory fitness, muscular strength, and motor fitness) with SRH at baseline (cross-sectional) and two years later (longitudinal) in youth.

## 2. Methods

### 2.1. Study Design, Settings, and Participants

Participants were enrolled in the UP&DOWN study [21], which is a longitudinal study designed to assess the impact over time of physical activity and sedentary behaviour on health indicators, as well as to identify the psycho-environmental and genetic determinants of physical activity in a convenience sample of Spanish children and adolescents. A total of 24 primary schools and 46 secondary schools from Cadiz and Madrid regions (Spain), respectively, were invited to participate. Twenty-three primary schools and 22 secondary schools accepted the invitation, however, four secondary schools were excluded for logistical reasons. All children from 1st and 4th grades and adolescents from 7th and 10th grades were invited to participate.

The UP&DOWN study includes a convenience sample of 2225 children and adolescents aged 6–18 years. Participants aged <8 years (*n* = 847) did not complete self-reported measures as self-reported questionnaires for SRH doesn’t show validity for this range of age. In the present study, children and adolescents aged 8 to 17.9 years at baseline were included as long as they had complete data at baseline and follow-up on body mass index (BMI), pubertal status, SRH, cardiorespiratory fitness, muscular strength, and motor fitness. Hence, the present study analyzed 1378 children and adolescents (687 children) at baseline and 1108 children and adolescents (613 children), as 19.5% dropped out.

For recruitment, an invitation letter to participate in this study was sent to the headmasters or physical education teachers in each school. Once schools had accepted the invitation, a meeting with the headmasters was organized to explain the objective of the study and to ask for their participation in the project. In addition, parents of students received a brief flyer describing the study, an explanation of inclusion criteria, and an invitation to attend an information session at the school during which the study purpose was explained and written informed consent from parents/caregivers was obtained.

Baseline data were collected from September 2011 to June 2012, and follow-up data were collected from September 2013 to June 2014.

The study complied with the Declaration of Helsinki and protocols were approved by the Ethics Committee of the Hospital Puerta de Hierro (Madrid, Spain), the Bioethics Committee of the National Research Council (Madrid, Spain), and the Committee for Research Involving Human Subjects at University of Cádiz. Parents and school supervisors were informed by letter about the nature and purpose of the study, and written informed consent was provided.

### 2.2. Measurements

#### 2.2.1. Physical Characteristics

Individual-level covariates, including gender, age, BMI (Body Mass Index), and pubertal status were recorded. Weight and height were assessed following the protocols of the ALPHA (Assessing Levels of Physical Activity) health related fitness test battery for youth [22]. Weight and height were measured twice, and averages were recorded. BMI was calculated as weight in kilograms divided by height in meters squared (kg/m^2^). Pubertal status was self-reported according to the five stages defined by Tanner and Whitehouse, based on breast development and pubic hair in girls, and penis and scrotum development and pubic hair in boys [23].

#### 2.2.2. Self-Rated Health (SRH)

Subjective health was assessed by the classic SRH status item that consists of asking respondents to rate their health as excellent, good, fair, bad, or poor [21], which has shown good reliability and validity in children [13,14,24] and adolescents [10,11,24]. In children, the scientist in charge of this research tool, helped them one by one to clarify the question and ratings in order to minimize possible bias because immaturity [25].

#### 2.2.3. Physical Fitness

Physical fitness was assessed following the ALPHA health-related fitness test battery for youths [26,27]. All tests were performed in a single session.

Cardiorespiratory fitness was assessed by the 20 m shuttle-run test. The participants were required to run between two lines 20 m apart, while keeping pace with a pre-recorded audio CD. The initial speed was 8.5 km·h^−1^, which was increased by 0.5 km·h^−1^ each minute (1 min = one stage). Participants were instructed to run in a straight line, to pivot on completing a shuttle (20 m), and to pace themselves in accordance with the audio signals. The test was finished when the participant failed to reach the end lines concurrent with the audio signals on two consecutive occasions [28]. The test was performed once, always at the end of the sequence, and the last completed stage at which the subject dropped out was scored.

Muscular strength was assessed based on maximum handgrip strength (upper isometric muscular strength) and the standing long jump (lower body explosive strength) tests. A hand dynamometer with an adjustable grip was used (TKK 5101 Grip D; Takey, Tokyo, Japan) for the handgrip strength test [29,30]. The grip-span of the dynamometer was adjusted according to the hand size of the youth [30]. The participants squeezed the dynamometer gradually for at least 2 s alternatively with both hands and the elbow in full extension as described elsewhere [29]. The test was performed twice and the highest score in kilograms for each hand was recorded. The average score of the left and right hand was calculated and divided by body weight (WG) as a relative measurement of upper body isometric muscular strength (UIMS/WG). The standing long jump test was performed from a starting position immediately behind a line, standing with feet approximately shoulder width apart. The participant jumped as far forwards as possible on a non-slip hard surface. The test was performed twice and the best score was recorded in centimeters as a relative measurement of lower body explosive muscular strength [28].

Motor fitness was assessed with the 4 × 10 m shuttle run test of speed of movement, agility, and coordination. The participants were required to run back and forth between two parallel lines 10 m apart. They were asked to run as fast as possible from the starting line to the other line and they should pick up (the first time) or exchange (second and third time) a sponge that has earlier been placed behind the lines. The test was performed twice, and the fastest time was recorded in seconds [28]. As the motor fitness score is inversely related to high physical fitness, it was first multiplied by −1, and a higher score indicates better motor fitness.

Additionally, muscular strength and a global physical fitness scores were calculated by summing up the z-scores ((value-mean)/standard deviation) of the subsequent physical fitness test scores. The muscular strength score was calculated by summing up the z-scores of handgrip strength/WG and standing long jump tests, while for the global fitness scores, the 20 m shuttle-run, handgrip strength/WG, standing long jump, and 4 × 10 m shuttle run z-scores were summed up. Z-scores were calculated by age group (children vs. adolescents) and by gender (boys vs. girls). Participants were categorized as children or adolescents using the median of the age as a cut-off point (children: 8 to 11.9 years age; adolescents: 12 to 17.9 years age).

### 2.3. Statistical Analysis

Descriptive sample characteristics are presented as a mean (standard deviation) or as number (percentage). Differences between baseline and 2-year follow-up within each age group were analyzed using a one-way ANOVA for continuous variables, and Chi-squared test for categorical variables. Preliminary analyses showed significant interaction for age*physical fitness tests but not for gender*physical fitness tests in relation to SRH. Thus, following analyses were performed splitting the sample by age groups (i.e., children and adolescents).

Cumulative link models [31] were fitted to analyze the association between physical fitness at baseline and SRH at baseline and 2-year follow-up. Five different models were fitted (two for cross-sectional and three for follow-up analyses). Cross-sectional models included SRH at baseline as a dependent variable, and one of each component of the physical fitness, muscular strength score, and the global physical fitness separately (model 1), or all physical fitness tests simultaneously (model 2), as independent variables. Gender, BMI, and pubertal status were added as covariates in model 1 and 2. Model 3 was fitted to assess the association of each component of the physical fitness, muscular strength score, and the global physical fitness separately at baseline with the SRH at 2-year follow-up. Model 4 was fitted to analyze which physical fitness tests had the strongest association when all were included simultaneously with SRH at 2-year follow-up. Finally, model 5 included the difference between baseline and 2-year follow up for SRH (ΔSRH) as the dependent variable and difference between baseline and 2-year follow for global physical fitness (Δ global physical fitness) as the independent variable. Models 3, 4, and 5 were adjusted by SRH at baseline, gender, BMI, and pubertal status at 2-year follow-up.

BMI was removed as covariate when UIMS/WG was included as independent variable in all the analyses given that the effect of body weight is already being counted.

Differences in the SRH at follow-up between cardiorespiratory fitness score groups (low vs. high) at baseline and among categories of changes in cardiorespiratory fitness were analyzed using an ANCOVA with SRH at baseline, gender, BMI, and pubertal status at 2-year follow-up as covariates. Cardiorespiratory fitness score at baseline was categorized as low or high according to the cut-off points proposed by Castro-Piñero et al. [8] for children and by Ruiz et al. [32] for adolescents. Moreover, participants were classified into four groups of change in cardiorespiratory fitness level: “persistent low” (low cardiorespiratory fitness at both baseline and follow-up), “decreasing” (drop from high to low cardiorespiratory fitness), “persistent high” (high cardiorespiratory fitness at both baseline and follow-up), and “increasing” (increased cardiorespiratory fitness from low to high at follow-up). Post-hoc comparisons were performed with the package emmeans [33]. Analyses were performed using R Statistical Software [34]. For all the models fitted multicollinearity analysis reported a VIF < 10 for all variables, while residual analysis showed normality of residuals for physical fitness variables. The level of significance was set at *p* < 0.05.

## 3. Results

Table 1 presents descriptive characteristics of the study sample. Cardiorespiratory fitness, UIMS/WG and lower body explosive muscular strength scores were significantly increased from baseline to follow-up in children and adolescents (all *p* < 0.001). Conversely, SRH decreased significantly from baseline to follow-up in adolescents (*p* < 0.001).

Table 2 shows the association of physical fitness with SRH at baseline by age groups. Baseline analyses showed that cardiorespiratory fitness, UIMS/WG, muscular strength score, and global physical fitness were positively associated with SRH in children (model 1; OR, 1.048; 95% CI, 1.020–1.076; OR, 18.921; 95% CI, 3.47–104.355; OR, 1.213; 95% CI, 1.117–1.319; and OR, 1.170; 95% CI, 1.081–1.266, respectively; all *p* < 0.001) and adolescents (OR, 1.057; 95% CI, 1.037–1.076; OR, 5.707; 95% CI, 1.122–29.205; OR, 1.169; 95% CI, 1.070–1.278, and OR, 1.154 95% CI, 1.100–1.210, respectively; all *p* < 0.001). Additionally, motor fitness was positively associated with SRH only in adolescents (OR, 1.192; 95% CI, 1.066–1.309; *p* = 0.007). When all the physical fitness variables were added simultaneously (model 2), only cardiorespiratory fitness was independently associated with SRH in children (OR, 1.059; 95% CI, 1.029–1.090) and adolescents (OR, 1.073; 95%CI, 1.050–1.097) with both *p* < 0.001.

The 2-year follow-up analyses (Table 3) showed that cardiorespiratory fitness, and global physical fitness at baseline were positively associated with SRH at follow-up in children (model 3; OR, 1.056; 95% CI, 1.023–1.091; *p* < 0.001; and OR, 1.082; 95% CI, 1.031–1.136, *p* < 0.01; respectively). When all the physical fitness variables at baseline were added simultaneously (model 4), only cardiorespiratory fitness was independently associated with SRH at follow-up in children (OR, 1.075; 95% CI, 1.040–1.112; *p* < 0.001) and adolescents (OR, 1.043; 95% CI, 1.014–1.074; *p* < 0.01). Finally, Δ global physical fitness was not associated with ΔSRH in either children or adolescents.

ANCOVA analysis of cardiorespiratory fitness at baseline with SRH at 2-year follow-up is shown in Figure 1. Children with a high level of cardiorespiratory fitness at baseline reported a significantly higher level of SRH at 2-year follow-up (*p* < 0.01) than their peers with lower level of cardiorespiratory fitness. However, children and adolescents with a high level of cardiorespiratory fitness at baseline reported a significantly higher level of SRH at 2-year follow-up when they were grouped as low or high according to the median (data not shown).

Figure 2 shows the association between cardiorespiratory fitness-change (persistent low, decreasing, persistent high, and increasing) categories with SHR score at follow-up. Children categorized as “persistent high” reported higher SRH at 2-year follow-up than those from “persistent low” and “decreasing” groups (both contrasts *p* < 0.01). Similarly, adolescents in the “persistent high” group reported higher SRH at 2-year follow-up than those classified as “persistent low” (*p* < 0.05).

## 4. Discussion

The main findings of the present study indicate that: Cardiorespiratory fitness, UIMS/WG, muscular strength score, and global physical fitness were positively associated with SRH in children and adolescents, and motor fitness was positively associated with SRH only in adolescents at baseline; cardiorespiratory fitness and global physical fitness at baseline were positively associated with SRH in children 2 years later; only cardiorespiratory fitness at baseline was independently associated with SRH in children and adolescents at baseline and 2-year follow-up; and a high level of cardiorespiratory fitness at baseline or maintaining high levels of cardiorespiratory fitness from the baseline to 2-year follow-up were associated with a higher level of SRH at 2-year follow-up in children and adolescents. These results suggest the importance of cardiorespiratory fitness as a predictor of SRH at present and future, making it of a great value to public health. Interventions programs to increase physical activity levels to enhance cardiorespiratory fitness level would be an important objective for schools and public health institutions to improve SRH level of the youth population.

Results of SRH are quite different in children compared to adolescents. These results are line with previous studies [12,13,24]. This can be due to adolescence being a period of life characterized by many physiological and psychological changes, [6] thus, adolescents tend to be more volatile emotionally than children and often experience extremes of mood [35] that might influence their SRH.

Collectively, the findings of previous cross-sectional studies are partially consistent with our results and reveal positive associations between cardiorespiratory fitness, [10,11,12,14] UIMS/WG, [13,15], and global physical fitness [24] with SRH in children and adolescents. In contrast, other studies did not find an association between cardiorespiratory fitness [12] and muscular strength (assessed by absolute UIMS) [14] with SRH in children. On the other hand, we also found that motor fitness was positively associated with SRH only in adolescents. Motor fitness implies the acquisition and subsequent refinement of novel combinations of movement sequences [36] and might occur when children are not able to reach a minimum level of motor fitness, which prevents the possibility of finding a positive association with SRH. To the best of our knowledge, no previous studies have investigated the association of motor fitness with SRH in youth. Only one in aged adults [37] also found a positive association. Therefore, additional studies are needed in youth to contrast or confirm present findings.

In addition, we examined whether components of physical fitness (cardiorespiratory fitness, muscular strength, and motor fitness) at baseline were associated with a better SRH at 2-year follow-up. We observed that cardiorespiratory fitness, muscular strength score, and global physical fitness were positively associated with SRH only in children, two years later. A 20-year follow-up study in Spanish older adults showed that those who had worse self-reported physical fitness status tended to have a relatively lower SRH [19]. Moreover, Starr et al. [38] reported that absolute upper body muscular strength was positively associated with SRH at 5-year follow-up in elderly people. As far as we know, there are no available longitudinal studies investigating the association between physical fitness components with SRH in a youth population. Whether muscular strength and motor fitness are associated with SRH in youth remains unknown, and future longitudinal studies are needed.

Importantly, we studied the possible independent association of the three physical fitness components included simultaneously (cardiorespiratory fitness, muscular strength, and motor fitness) with SRH at baseline (cross-sectional) and at follow-up (longitudinal). Only cardiorespiratory fitness at baseline was independently associated with SRH in both age groups at baseline and follow-up. The results of a positive association between global physical fitness and SRH might be mediated by cardiorespiratory fitness. The mechanism explaining these positive results only with cardiorespiratory fitness might be in part because cardiorespiratory fitness induces cortical angiogenesis, increases blood flow, and elevates the expression of brain-derived neurotrophic factor proteins (BDNF) [36]. Those phenomenon might improve psychological well-being [39] and SRH levels, as SRH may extend beyond symptoms and be a somatic expression of life distress. [40] The lack of studies analyzing the interdependent influence of the components of physical fitness with SRH in youths prevents the possibility of comparing present results with others.

Recently, two studies provided pooled cardiorespiratory fitness cut-off points in children [8] and adolescents [32] that could serve as a standard level to meet health-related cardiorespiratory fitness. In our study we used the proposed cut-off points to test whether they could be used as a predictor of SRH level. We found that children who met cardiorespiratory fitness cut-off points at baseline were more likely to have a higher SRH level at follow-up than those who met cardiorespiratory fitness below cut-off points. However, we did not find differences in adolescents. Despite the significant discriminating accuracy of these cut-off points, [8] not finding an association in adolescents might be due to the high sensitivity of this cut-off points in youth and because in our study only 26% of adolescents met these values. Indeed, when we established cardiorespiratory fitness groups (low and high) according to the median (36.39 and 31.83 for children boys and girls, respectively; 34.55 and 33.03 for adolescents boys and girls, respectively), we found that children and adolescents with a high level of cardiorespiratory fitness at baseline reported significant higher level of SRH at 2-year follow-up (*p* < 0.01 for children; *p* < 0.05 for adolescents) than their peers with a lower level of cardiorespiratory fitness. Additionally, we observed that children and adolescents who maintained their cardiorespiratory fitness with similar levels or above the cut-off points had higher levels of SHR scores than their counterparts who maintained their cardiorespiratory fitness levels below cut-off points; or than their counterparts who decreased their cardiorespiratory fitness levels below cut-off points with respect to children. However, when we analyzed whether changes in global physical fitness determined changes in SHR at 2-year follow-up, we did not find any association in either children or adolescents.

Levels of cardiorespiratory fitness tend to track from childhood to adolescence [41,42] and adulthood [43,44]. Our results suggest that youth health interventions should focus not only on medical resources, but also on the development of strategies to promote physical activity and exercise training for improving level of physical fitness, specifically cardiorespiratory fitness, during childhood and adolescence. Furthermore, SRH is strong correlated with mortality and objective health status. [2,45] Hence, it deserves more research attention as a health indicator among youth and how the determinants change over time. Health professionals should pay attention to patients’ SRH and assess cardiorespiratory fitness level as well for risk stratification.

### Strengths and limitations

Limitations of the present study its use of a convenience sample, which limits the generalizability of our findings across the population. Another limitation is the fact that SRH is based on a single (yet validated and widely used) question item [2,46,47]. On the other hand, the strengths of this study are the relatively large and heterogeneous sample focused on two age groups (children and adolescents). Furthermore, a longitudinal design including the independent and combined study of the associations of the three components of physical fitness (cardiorespiratory fitness, muscular strength, and motor fitness) with SRH is used, taking into account several confounding factors in the statistical analysis.

## 5. Conclusions

Cardiorespiratory fitness was independently and positively associated with SRH in children (8–11.9 years age) and adolescents (12–17.9 years age) at baseline and 2-year follow-up. A high level of cardiorespiratory fitness or maintaining high levels of cardiorespiratory fitness from the baseline to 2-year follow-up was associated with a higher level of SRH at 2-year follow-up in children and adolescents. These findings emphasize the importance of cardiorespiratory fitness as strong predictor of present and future SRH in youth. Interventions programs to enhance cardiorespiratory fitness level from early ages are urgently needed for present and future youth health, and as a cost-effective strategy.

## Figures and Tables

**Figure 1 ijerph-17-02413-f001:**
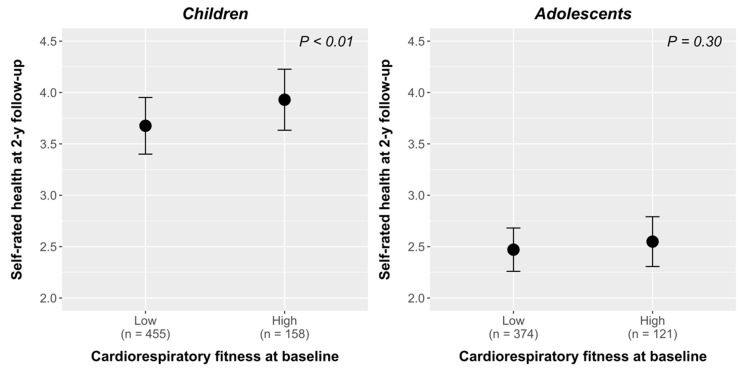
Self-rated health (SRH) at follow-up according to cardiorespiratory fitness level (low or high) in children and adolescents. The analysis was adjusted by levels of SRH at baseline, gender, BMI, and pubertal status, at 2-year follow-up. Significance differences were found between cardiorespiratory fitness levels for children *(p <* 0.01) using a post-hoc analysis with Bonferroni correction.

**Figure 2 ijerph-17-02413-f002:**
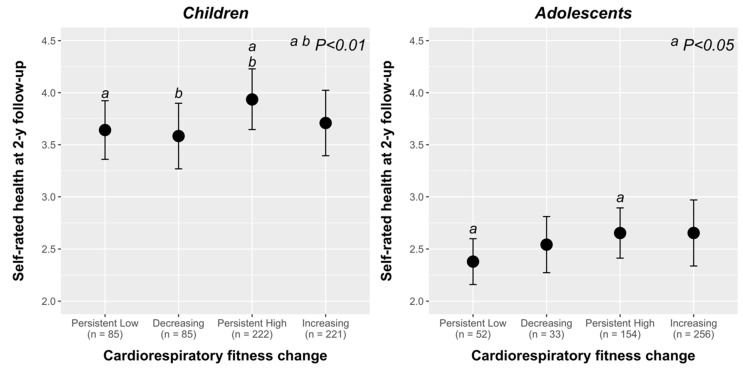
SRH at follow-up according to cardiorespiratory fitness change categories (persistent low, decreasing, persistent high, or increasing) in children and adolescents. Gender, BMI, and pubertal status at 2-year follow-up and SRH at baseline were included as covariates in the model. Categories with the same letter indicate a significant pairwise difference using a post-hoc analysis with Bonferroni correction.

**Table 1 ijerph-17-02413-t001:** Baseline and 2-year follow-up descriptive characteristics of the sample by age group.

				Children					Adolescents				
Variables				Baseline		2-year Follow-Up			Baseline		2-year Follow-Up		
				(*n* = 687)		(*n* = 613)		*p*	(*n* = 691)		(*n* = 495)		*p*
**Physical characteristics**													
Age (years)				10.63	(1.37)	12.59	(1.33)	**<0.001**	14.65	1.49	16.34	(1.39)	**<0.001**
Weight (kg)				41.33	(11.17)	50.18	(11.03)	**<0.001**	56.69	11.88	61.31	(11.72)	**<0.001**
Height (cm)				144.22	(10.44)	155.6	(10.45)	**<0.001**	162.73	9.38	167.5	(8.51)	**<0.001**
Weight status (*n*, %)													
		Underweight		21	(3)	24	(4)	**<0.001**	22	(3)	19	(3)	**<0.001**
		Normal weight		357	(52)	355	(58)		516	(75)	371	(75)	
		Overweight		235	(34)	178	(29)		116	(17)	86	(17)	
		Obese		74	(11)	56	(9)		37	(5)	19	(3)	
Pubertal status (*n*, %)													
		1		98	(14)	4	(0.6)	**<0.001**	36	(5)	-	-	**<0.001**
		2		467	(68)	138	(22)		201	(29)	3	(0.6)	
		3		114	(17)	270	(44)		311	(46)	67	(13)	
		4		8	(1)	142	(23)		137	(20)	218	(44)	
		5		-	-	59	(9)		-	-	207	(42)	
**Self-reported health** (*n*, %)				3.99	(0.81)	4.00	(0.82)	0.867	3.74	(0.81)	2.64	(0.80)	**<0.001**
		1		2	(0.3)	2	(0.3)	**<0.001**	2	(0.3)	-	-	**<0.001**
		2		17	(2)	13	(2)		32	(5)	31	(6)	
		3		166	(24)	153	(25)		232	(34)	185	(37)	
		4		301	(44)	258	(42)		306	(44)	209	(42)	
		5		201	(29)	187	(31)		119	(17)	70	(14)	
**Physical fitness**													
Cardiorespiratory fitness (stage)				3.69	(1.84)	4.93	(0.99)	**<0.001**	5.78	(2.53)	6.32	(1.00)	**<0.001**
Upper isometric muscular strength (kg)				16.91	(4.74)	22.27	(7.05)	**<0.001**	27.67	(7.77)	32.26	(8.06)	**<0.001**
UIMS/WG				0.41	(0.08)	0.44	(0.10)	**<0.001**	0.49	(0.10)	0.53	(0.10)	**<0.001**
Lower body explosive strength (cm)				130.35	(23.39)	145.0	(27.51)	**<0.001**	164.06	(33.51)	174.7	(36.84)	**<0.001**
Motor fitness (s)				13.27	(1.15)	12.43	(0.98)	0.454	12.07	(1.14)	11.75	(1.01)	0.395
Muscular strength score (z-score)				−0.03	(1.73)	−0.01	(1.69)	0.895	0.06	(1.70)	0.02	(1.66)	0.733
Global physical fitness (z-score)				−0.07	(3.22)	−0.02	(1.92)	0.798	0.19	(3.08)	0.03	(1.77)	0316

Values represent mean (standard deviation) unless otherwise indicated. UIMS/WG indicates upper isometric muscular strength divided by body weight. Significant differences between baseline and 2-year follow-up are highlighted in bold.

**Table 2 ijerph-17-02413-t002:** Cumulative odd ratios and 95% confidence intervals for having high self-rated health according to physical fitness at baseline.

	Children (*n* = 687)			Adolescents (*n* = 691)		
	OR	95% CI	*p*	OR	95% CI	*p*
*Model 1*						
Pubertal status	1.068	0.834–1.369	0.601	1.050	0.873–1.265	0.603
Gender	1.156	0.847–1.580	0.360	0.844	0.581–1.226	0.373
BMI	**0.942**	**0.898–0.988**	**0.014**	**0.919**	**0.874–0.966**	**<0.001**
Cardiorespiratory fitness	**1.048**	**1.020–1.076**	**<0.001**	**1.057**	**1.037–1.076**	**<0.001**
Pubertal status	0.964	0.757–1.227	0.766	**0.903**	**0.755–1.078**	**0.260**
Gender	0.995	0.748–1.325	0.975	**0.490**	**0.357–0.672**	**<0.001**
UIMS/WG	**18.921**	**3.471–104.355**	**<0.001**	**5.707**	**1.122–29.205**	**0.036**
Pubertal status	1.061	0.828–1.359	0.641	1.032	0.852–1.252	0.747
Gender	0.946	0.707–1.267	0.710	**0.515**	**0.360–0.736**	**<0.001**
BMI	**0.911**	**0.872–0.952**	**<0.001**	**0.882**	**0.841–0.925**	**<0.001**
Lower body explosive strength	1.002	0.996–1.008	0.535	1.004	0.999–1.010	0.112
Pubertal status	1.057	0.825–1.354	0.661	1.004	0.828–1.217	0.969
Gender	0.967	0.719–1.299	0.823	**0.554**	**0.393–0.778**	**<0.001**
BMI	**0.915**	**0.874–0.956**	**<0.001**	**0.889**	**0.847–0.932**	**<0.001**
Motor fitness	1.067	0.933–1.184	0.313	**1.192**	**1.066–1.309**	**0.007**
Pubertal status	0.976	0.766–1.243	0.841	0.899	0.753–1.073	0.237
Gender	0.982	0.740–1.303	0.900	**0.482**	**0.356–0.650**	**<0.001**
Muscular strength score	**1.213**	**1.117–1.319**	**<0.001**	**1.169**	**1.070–1.278**	**<0.001**
Pubertal status	0.980	0.769–1.249	0.872	0.898	0.752–1.073	0.236
Gender	0.959	0.724–1.272	0.773	**0.472**	**0.350–0.635**	**<0.001**
Global physical fitness	**1.170**	**1.081–1.266**	**<0.001**	**1.154**	**1.100–1.210**	**<0.001**
*Model 2*						
Pubertal status	1.012	0.794–1.292	0.921	0.967	0.802–1.166	0.723
Gender	1.233	0.902–1.688	0.189	0.948	0.646–1.391	0.785
Cardiorespiratory fitness	**1.059**	**1.029–1.090**	**<0.001**	**1.073**	**1.050–1.097**	**<0.001**
UIMS/WG	5.993	0.670–54.072	0.109	0.487	0.063–3.757	0.490
Lower body explosive strength	1.000	0.991–1.009	0.954	0.996	0.988–1.003	0.342
Motor fitness	0.979	0.804–1.192	0.829	1.180	0.944–1.475	0.147

OR: Cumulative odd ratios; CI: confidence intervals; UIMS/WG: Upper isometric muscular strength divided by body weight. Model 1: each model included one fitness test score separately. Covariates: pubertal status, gender, and BMI. Model 2: All physical fitness test scores were included simultaneously. Covariates: pubertal status, gender, and BMI. BMI was removed as covariate in the models that included UIMS/WG as independent variable. Significant values are indicated in bold.

**Table 3 ijerph-17-02413-t003:** Cumulative odd ratios and 95% confidence intervals for having high self-rated health at 2-year follow-up according to physical fitness at baseline.

	Children (*n* = 613)			Adolescents (*n* = 496)		
	OR	95% CI	*p*	OR	95% CI	*p*
*Model 3*						
Pubertal status	0.854	0.714–1.019	0.080	0.983	0.752–1.285	0.900
Gender	0.873	0.627–1.217	0.423	0.603	**0.380–0.953**	**0.031**
BMI	**0.954**	**0.910–0.999**	**0.044**	0.961	0.907–1.018	0.175
SRH baseline	**1.910**	**1.562–2.341**	**<0.001**	3.652	**2.806–4.783**	**<0.001**
Cardiorespiratory fitness	**1.056**	**1.023–1.091**	**<0.001**	1.024	0.999–1.051	0.058
Pubertal status	**0.746**	**0.629–0.885**	**<0.001**	0.942	0.724–1.225	0.656
Gender	**0.705**	**0.520–0.956**	**0.025**	**0.477**	**0.318–0.713**	**<0.001**
SRH baseline	**2.032**	**1.663–2.488**	**<0.001**	**3.980**	**3.082–5.176**	**<0.001**
UIMS/WG	2.154	0.344–13.519	0.412	0.981	0.135–7.164	0.985
Pubertal status	**0.782**	**0.643–0.949**	**0.013**	1.027	0.782–1.350	0.847
Gender	**0.726**	**0.530–0.992**	**0.045**	**0.412**	**0.266–0.633**	**<0.001**
BMI	**0.933**	**0.891–0.976**	**0.002**	**0.942**	**0.890–0.998**	**0.042**
SRH baseline	**1.972**	**1.615–1.012**	**<0.001**	**3.871**	**2.985–5.055**	**<0.001**
Lower body explosive strength	1.004	0.996–1.012	0.326	0.996	0.989–1.002	0.164
Pubertal status	**0.810**	**0.673–0.974**	**0.025**	1.001	0.763–1.314	0.993
Gender	**0.703**	**0.512–0.964**	**0.029**	**0.452**	**0.296–0.687**	**<0.001**
BMI	**0.928**	**0.886–0.971**	**0.001**	0.947	0.894–1.002	0.061
SRH baseline	**1.981**	**1.622–2.427**	**<0.001**	**3.857**	**2.972–5.039**	**<0.001**
Motor fitness	1.022	0.873–1.196	0.785	0.952	0.794–1.141	0.592
Pubertal status	**0.744**	**0.627–0.881**	**<0.001**	0.934	0.717–1.218	0.615
Gender	**0.712**	**0.526–0.962**	**0.027**	**0.491**	**0.332–0.724**	**<0.001**
SRH baseline	**1.980**	**1.619–2.427**	**<0.001**	**4.040**	**3.118–5.271**	**<0.001**
Muscular strength score	1.110	1.105–1.213	0.052	1.012	0.911–1.125	0.822
Pubertal status	**0.797**	**0.667–0.950**	**0.011**	0.956	0.729–1.254	0.747
Gender	**0.691**	**0.511–0.934**	**0.016**	**0.461**	**0.313–0.678**	**<0.001**
SRH baseline	**1.901**	**1.554–2.333**	**<0.001**	**3.715**	**2.860–4.856**	**<0.001**
Global physical fitness	**1.086**	**1.026–1.150**	**0.004**	1.053	0.990–1.119	0.101
*Model 4*						
Pubertal status	**0.808**	**0.665–0.980**	**0.031**	1.016	0.775–1.333	0.901
Gender	0.946	0.677–1.323	0.747	**0.566**	**0.355–0.898**	**0.016**
SRH baseline	**1.943**	**1.588–2.384**	**<0.001**	**3.720**	**2.861–4.870**	**<0.001**
Cardiorespiratory fitness	**1.075**	**1.040–1.112**	**<0.001**	**1.043**	**1.014–1.074**	**0.003**
UIMS/WG	0.214	0.020–2.213	0.196	1.665	0.129–21.579	0.696
Lower body explosive strength	1.008	0.998–1.018	0.134	0.991	0.981–1.001	0.068
Motor fitness	0.917	0.740–1.137	0.431	1.003	0.765–1.313	0.985
*Model 5*						
Pubertal status	**1.339**	**1.124–1.596**	**0.001**	1.071	0.818–1.404	0.617
Gender	**1.494**	**1.099–2.036**	**0.010**	**2.145**	**1.445–3.200**	**<0.001**
SRH baseline	**5.730**	**4.535–7.296**	**<0.001**	**4.294**	**3.296–5.646**	**<0.001**
Δ Global physical fitness	0.942	0.883–1.006	0.076	1.002	0.922–1.088	0.965

OR: Cumulative odd ratios; CI: confidence intervals; UIMS/WG: upper isometric muscular strength divided by body weight; SRH: self-reported health; Δ: value at two year minus value at baseline. Model 3: each model included one fitness test score at baseline separately. Covariates: pubertal status, gender, and BMI after 2-year follow up and levels of SRH at baseline. Model 4: All physical fitness test scores at baseline were included simultaneously. Covariates: pubertal status, gender, and BMI after 2-year follow up and levels of SRH at baseline. Model 5: dependent variable for this model is ΔSRH while the independent variables was Δ Global physical fitness. Covariates: pubertal status, and gender after 2-year follow up and levels of SRH at baseline. BMI was removed as covariate in the models that included UIMS/WG as independent variable. Significant values are indicated in bold.

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
