# Peer review of "Physical Fitness and Self-Rated Health in Children and Adolescents: Cross-Sectional and Longitudinal Study"

_ijerph, 2020, doi:10.3390/ijerph17072413_

Round 1
Reviewer 1 Report
Comments
This is an interesting piece of work and worth publishing. This is one of the occasions where the study is really interesting but unfortunately the authors do almost everything that is within their power to ruin the result. However, I feel that Spanish children and children overall would deserve these results. Therefore, I am encouraging you to put some hours to improve the contents and think carefully all intermediate steps. I am trying to give some hints below, but as you may go along, more questions may arise. Overall, I would encourage in incorporating more statistical knowledge into the study analysis and possibly among authors.
General comments.
The study design need to be carefully described, possibly in a supplementary file. How were persons selected? How were schools selected? Did you interview 8 year olds or their care givers or both? Please compare to e.g. school statistics.
The fact that SRH is a good indicator does not mean that it is a good indicator in young children or adolescents. They may have different taking on their health, also their care givers. Of your references, the youngest had a mean age of around 17 years (Randy et al., 2009). You should try to include some studies involving children and challenges in obtaining results. Also the SRH is quite different in youngers vs. olders (e.g. Figure 1), any explanations? As a personal experience, I may state that interviewing young children about their food consumption habits (in schools) is useful and usually better data is obtained from children than their care givers (especially fathers do not often have a clue what their children eat, wear or have as health issues), but the understanding of “health” may be vague among young children.
It would be nice to see the results of the both age groups combined. What was the rationale of splitting into two, why not three? If the result remains, it would be nice just to report without age groups. You could also include age as controlling variable like gender, BMI and puberty score. Otherwise, try to give in the end recommendations for both groups separately.
You could also try to make one additional model, based on the difference between baseline and 2-year follow up, ie. ΔSRH ~ Δglobal physical fitness, where Δ = (value at two year minus value at baseline). If the adjustment factors would be added, this would be more useful possibly than currently presented figures.
What is the correlation between SRH and BMI, you may end up having the same variable essentially as outcome and explanatory (not allowed)? [Chronic illnesses are rare in children, and in Europe infectious diseases are low due to efficient vaccination programs, ie. BMI may explain large proportion of SRH]. If SRH and BMI are highly correlated, that is a useful result as such and BMI could be used as a proxy of SRH in the future studies. Please report.
As it is likely that physical fitness factors are highly correlated, you should carry out formal testing for VIF (variation inflation factor) or other methods to assess better the correlation. The variable with highest OR/lowest p-value is not necessarily the best explanation, rather use AIC (please report) or similar. I would also propose to only include variables separately, and then summed z-scores (after possible normalization), not all in lump. Correlation may produce odd OR/p-value combinations. Did you consider interaction terms? Also please include the results of gender, BMI and puberty scores in adjusted models.
Did you consider your link function for the outcome variable? Please give rationale.
Did your z-scores follow normal distribution (please test and report), if not, you might like to consider e.g. log-transformed values, especially for summing up.
Did you study any model diagnostics or normality of residuals. Please report.
I do not understand “Covariates: gender, BMI and pubertal status at 2-year follow up and levels of SRH at baseline”.
How is the situation in children: “Furthermore, SRH is strong correlated with mortality and objective health status. [4, 51]”
If e.g. AIC values are lower for global physical fitness than for cardiorespiratory fitness alone, you might like to recommend overall improvement in physical fitness important.
Minor comments:
l. 54-56 and 62-64. Please check logic.
Table 1. weight status and SRH between baseline and 2-year follow up are (almost) identical in youngers and olders, yet you get a p-value <0.001?
Wording “youngers” and “olders” does not sound proper English.
Table 1 and Table 2. Please name physical variables as cardiorespiratory fitness, muscular strength and motor fitness rather than testing method.
Author Response
ANSWER TO THE REVIEWERS’ COMMENTS
REVIEWER -1-
Comment
This is an interesting piece of work and worth publishing. This is one of the occasions where the study is really interesting but unfortunately the authors do almost everything that is within their power to ruin the result. However, I feel that Spanish children and children overall would deserve these results. Therefore, I am encouraging you to put some hours to improve the contents and think carefully all intermediate steps. I am trying to give some hints below, but as you may go along, more questions may arise. Overall, I would encourage in incorporating more statistical knowledge into the study analysis and possibly among authors.
Answer
Please, find a revised version of our manuscript entitled “Physical fitness and self-rated health in youth: cross-sectional and longitudinal study” (before: “Cardiorespiratory fitness is a predictor of present and future self-rated health in youth”). We would like to thank the associate Editor and the Reviewers for their time and the assessment of our manuscript and for providing us with this opportunity to improve the quality of our paper based on their constructive feedback.
An itemized point-by-point response to the reviewers’ comments is presented below. We have considered all of the Reviewers’ suggestions, and have either incorporated them into the revised manuscript or offered our rationale for not doing so. Changes to the original manuscript are highlighted directly in the text using a yellow font.
The authors are really delighted for having the opportunity of publishing with your journal.
General comments.
Comment
The study design need to be carefully described, possibly in a supplementary file. How were persons selected? How were schools selected? Did you interview 8 year olds or their care givers or both? Please compare to e.g. school statistics. The fact that SRH is a good indicator does not mean that it is a good indicator in young children or adolescents. They may have different taking on their health, also their care givers. Of your references, the youngest had a mean age of around 17 years (Randy et al., 2009). You should try to include some studies involving children and challenges in obtaining results. Also the SRH is quite different in youngers vs. olders (e.g. Figure 1), any explanations? As a personal experience, I may state that interviewing young children about their food consumption habits (in schools) is useful and usually better data is obtained from children than their care givers (especially fathers do not often have a clue what their children eat, wear or have as health issues), but the understanding of “health” may be vague among young children.
Answer
The authors acknowledge the Reviewer’s suggestions. We do apologize for the apparently biased description of the methodology.
For the feasibility of our study and for the follow-up study, we followed a strict standardization of the fieldwork. Considering the suggestion of the reviewer to improve the clarity of the methods section. We have provided more information about the study design in the manuscript about selection procedures, the number of eligible schools, how many schools agreed to participate and the number of students agreed to participate in the UP&DOWN study. (See Study design, settings and participants, page 3, lines 86-93).
Moreover, related to reviewer´s question about interview SRH in children. Although there is always bias associated with self-reported measures, particularly in children, we tried to minimize this through:
- Methodological coordination and harmonization workshops previous to the research were held where the entire scientist taken part in.
- The use of standardized procedures and validated questionnaires used in previous studies in children [1-3] and adolescents [3-5]. These references have been modified in the manuscript to support this information. (See measurements: self rated health (SRH)- section, pages 4, line 128)
Furthermore, children and adolescents self-reported their subjective health rated in a simple and significative scale as: “excellent, good, fair, bad or poor” rated from “1 to 5” easy to understand for schoolchildren. In children, the scientist in charge of this research tool, helped them one by one to clarify this question and rates in order to minimize possible bias because immaturity, making our estimations conservative. [6] In addition, for the current study, participants aged <8 years did not complete self-reported measures about SRH, because there are not validity questionnaires for this range of age. Thus, they were not taking into account either in other papers of UP&DOWN study [7] were self-reported measures were required and not validity questionnaires were available for this range of age. (See Measurements (SRH)- section, page 4, lines 128-130) and Study design, settings and participants - section, page 3, lines 95-96)
On other hand, as reviewer suggest, results of SRH are quite different in youngers (children) vs. olders (adolescents). These results are line with previous studies. [1, 3, 8] This can be due because adolescence is a period of life characterized by many physiological and psychological change, [9] thus, adolescents tend to be more volatile emotionally than children and often experience extremes of mood [10] which might influence consequently their subjective evaluation of their health (SRH). (See Discussion- section, page 13, lines 289-293)
Comment
It would be nice to see the results of the both age groups combined. What was the rationale of splitting into two, why not three? If the result remains, it would be nice just to report without age groups. You could also include age as controlling variable like gender, BMI and puberty score. Otherwise, try to give in the end recommendations for both groups separately.
You could also try to make one additional model, based on the difference between baseline and 2-year follow up, ie. ΔSRH ~ Δglobal physical fitness, where Δ = (value at two year minus value at baseline). If the adjustment factors would be added, this would be more useful possibly than currently presented figures.
Answer
Thank you for the suggestion. The sample was splitting in two groups due to a significant interaction for age*physical fitness test score (age is a categorical variable with 2 levels: “children” and “adolescents” in the interaction analysis). Similar analysis has been performed previously with the same sample by García-Cervantes et al. [11] (See Statistical analysis - section, page 5, lines 174-176)
Following reviewer´s comment an additional model based on the difference between baseline and 2-year follow up has been added to tables 2 and 3 in the manuscript. Moreover, an additional model, (model 5) based on the difference between baseline and 2-year follow up has been performance. (See Tables- section, pages 8-11)
Comment
What is the correlation between SRH and BMI, you may end up having the same variable essentially as outcome and explanatory (not allowed)? [Chronic illnesses are rare in children, and in Europe infectious diseases are low due to efficient vaccination programs, ie. BMI may explain large proportion of SRH]. If SRH and BMI are highly correlated, that is a useful result as such and BMI could be used as a proxy of SRH in the future studies. Please report.
Answer
We want to thank the reviewer for this interesting suggestion.
Correlation between SRH and BMI is -0.231 at baseline and -0.240 at 2-year follow-up. In our research BMI was considered as covariate in model 1,2,3 and 4 and was removed as covariate when UIMS/WG (upper body isometric muscular strength/ body weight) was included as independent variable in all the analyses, given that the effect of body weight is already being counted.[12]
Association between BMI and SRH was not the purpose of our study. For sure, we will take into account your interesting suggestion, if we have the opportunity, in our next research. Currently, our team research is in progress to develop U&DOWN study phase II where participants in UP&DOWN-I- will be measured 6 years later. We hope to obtain more information about how SRH and physical fitness modified from childhood to adolescence.
Comment
As it is likely that physical fitness factors are highly correlated, you should carry out formal testing for VIF (variation inflation factor) or other methods to assess better the correlation. The variable with highest OR/lowest p-value is not necessarily the best explanation, rather use AIC (please report) or similar. I would also propose to only include variables separately, and then summed z-scores (after possible normalization), not all in lump. Correlation may produce odd OR/p-value combinations. Did you consider interaction terms? Also please include the results of gender, BMI and puberty scores in adjusted models.
Did you consider your link function for the outcome variable? Please give rationale.
Did your z-scores follow normal distribution (please test and report), if not, you might like to consider e.g. log-transformed values, especially for summing up.
Did you study any model diagnostics or normality of residuals. Please report.
I do not understand “Covariates: gender, BMI and pubertal status at 2-year follow up and levels of SRH at baseline”.
Answer
Thank you very much for these interesting suggestions.
Additional analyses have been performed following the recommendation proposed by the reviewer:
- Multicollinearity analysis showed no correlation among independent variable in any model (i.e. VIF < 10; table supplementary 1, see below).
- Results of model comparison using likelihood ratio test for cumulative link models are showed in table supplementary 2 (see below). We can observe that the significant independent variables are the same than using Wald test for hypothesis testing.
- The results of pubertal status, gender and BMI in adjusted are included in the manuscript for baseline (table 2) and 2-year follow-up models (table 3). (See Tables- section, pages 10-11)
- The link function for the outcome variable is the logit link giving a proportional odds model.
- Yes, z-scores and the global physical fitness score follow a normal distribution (p>0.05 for Shapiro-Wilk test).
- Residuals analysis for ordinal models have been analyzed and reported in figure 1 to 6 at the end of this document (see below). We can assume the normality of the residuals for all the fitted models.
- The sentence “Covariates: gender, BMI and pubertal status at 2-year follow up and levels of SRH at baseline” has been changed at the footnotes of the tables. (See Tables- section, pages 7-11)
Comment
How is the situation in children: “Furthermore, SRH is strong correlated with mortality and objective health status. [4, 51]”
If e.g. AIC values are lower for global physical fitness than for cardiorespiratory fitness alone, you might like to recommend overall improvement in physical fitness important.
Answer
Thank you very much to the reviewer for his constructive feedback. Previous research reported that SRH in youth (children and adolescents) is related to their overall sense of functioning [13] and health-related quality of life [14] but still deserves more research attention as a health indicator in youth.[15] On the other hand, numerous benefits of physical fitness´s components for physical and mental health are well known in youth. [9] Thus, the purpose of our study was to differentiate which components of physical fitness might be more strongly associated (independent or combined) with SRH in order to use them as a predictor of future SRH level.
Our results emphasize that physical fitness, particularly cardiorespiratory fitness, is as strong predictor of present and future SRH in children and adolescents. This is line with previous studies which observed that physical fitness is not only led to better subjective health but also to a better valuation of objective health in children and adolescents emphasizing the significance of physical fitness in health research.[3]
Additionally, taking into account reviewer´s suggestion, as we have named in previous comment a multicollinearity analysis was done and we observed that Δ global physical fitness was not associated with ΔSRH in both, children and adolescents. (See Results - section, page 6, lines 228-229)
To the best of our knowledge, this is the first longitudinal study who have investigated the independent and combined associations of components of physical fitness (cardiorespiratory fitness, muscular strength and motor fitness) with SRH in youth, in order to analyze the relative weight of each component on schoolchildren’s SRH. On the final results of our study, cardiorespiratory fitness is the only one component of physical fitness, which was independently and positively associated with SRH in children (8-11.9 years age) and adolescents (12-17.9 years age) at baseline and 2-years follow-up.
In line with our results due to the weight of cardiorespiratory fitness our suggestion, if the reviewer agree, is not only to develop preventive strategies by health-care institutions to effectively increase physical fitness level in children (very important), but to focus intervention particularly on cardiorespiratory fitness, in order to prevent low levels of SRH. Nevertheless, if the Reviewer still believe that we should keep just physical fitness, we would modify the conclusion.
Currently, as has been named in the previous comment my team research is in progress to develop U&DOWN study phase II where participants in UP&DOWN-I- will be measured 6 years later. We hope to obtain more information about how SRH and physical fitness modified from childhood to adolescence.
Minor comments:
Comment
- 54-56 and 62-64. Please check logic.
Answer
Thank you for the suggestion. Authors apologize for the misunderstanding. We wanted to seek there were previous cross-sectional studies but they didn´t investigate the independent association of components of physical fitness with SRH in youth (line 54-56, before). And there are no longitudinal studies investigating the independent and/or combined associations of components of physical fitness with SRH in youth. (Lines 62-64, before). According reviewer´s suggestion text has been modified for a better understanding. (See introduction-section, pages 2, lines 71-74).
Comment
Table 1. weight status and SRH between baseline and 2-year follow up are (almost) identical in youngers and olders, yet you get a p-value <0.001?
Answer
Thank you for the comment. We have repeated the analysis and the p-values reported are correct.
Comment
Wording “youngers” and “olders” does not sound proper English.
Answer
Thank you for the comment. As the reviewer suggest we have replaced the term “youngers” by children and “olders” by adolescents. Changes are highlighted directly in the whole text using a yellow font.
Comment
Table 1 and Table 2. Please name physical variables as cardiorespiratory fitness, muscular strength and motor fitness rather than testing method.
Answer
Thank you for the suggestion. Changes have been done in tables 1, 2 and 3. (See Tables- section, pages 7-11)
References
- Morales, P. F.; Sanchez-Lopez, M.; Moya-Martinez, P.; Garcia-Prieto, J. C.; Martinez-Andres, M.; Garcia, N. L.; Martinez-Vizcaino, V., Health-related quality of life, obesity, and fitness in schoolchildren: the Cuenca study. Qual Life Res 2013, 22, (7), 1515-23.
- Andersen, J. R.; Natvig, G. K.; Aadland, E.; Moe, V. F.; Kolotkin, R. L.; Anderssen, S. A.; Resaland, G. K., Associations between health-related quality of life, cardiorespiratory fitness, muscle strength, physical activity and waist circumference in 10-year-old children: the ASK study. Qual Life Res 2017, 26, (12), 3421-3428.
- Lämmle, L.; Woll, A.; Mensik, G.; Bös, K., Distal and proximal factors of health behaviors and their associations with health in children and adolescents. International Journal of envirommental research and public heatlh 2013,10, (7), 2944-78.
- Kantomaa, M. T.; Tammelin, T.; Ebeling, H.; Stamatakis, E.; Taanila, A., High levels of physical activity and cardiorespiratory fitness are associated with good self-rated health in adolescents. J Phys Act Health 2015, 12, (2), 266-72.
- Mota, J.; Santos, R. M.; Silva, P.; Aires, L.; Martins, C.; Vale, S., Associations between self-rated health with cardiorespiratory fitness and obesity status among adolescent girls. J Phys Act Health 2012, 9, (3), 378-81.
- Sanchez-Oliva, D.; Leech, R. M.; Grao-Cruces, A.; Esteban-Cornejo, I.; Padilla-Moledo, C.; Veiga, O. L.; Cabanas-Sanchez, V.; Castro-Pinero, J., Does modality matter? A latent profile and transition analysis of sedentary behaviours among school-aged youth: The UP&DOWN study. J Sports Sci 2020, 1-8.
- Castro-Pinero, J.; Carbonell-Baeza, A.; Martinez-Gomez, D.; Gomez-Martinez, S.; Cabanas-Sanchez, V.; Santiago, C.; Veses, A. M.; Bandres, F.; Gonzalez-Galo, A.; Gomez-Gallego, F.; Veiga, O. L.; Ruiz, J. R.; Marcos, A., Follow-up in healthy schoolchildren and in adolescents with Down syndrome: psycho-environmental and genetic determinants of physical activity and its impact on fitness, cardiovascular diseases, inflammatory biomarkers and mental health; the UP&DOWN study. BMC Public Health 2014, 14, 400.
- Padilla-Moledo, C.; Castro-Pinero, J.; Ortega, F. B.; Mora, J.; Marquez, S.; Sjostrom, M.; Ruiz, J. R., Positive health, cardiorespiratory fitness and fatness in children and adolescents. Eur J Public Health 2011, 22, (1), 52-6.
- Ortega, F. B.; Ruiz, J. R.; Castillo, M. J.; Sjostrom, M., Physical fitness in childhood and adolescence: a powerful marker of health. Int J Obes (Lond) 2008, 32, (1), 1-11.
- Arnett, J. J., Adolescent storm and stress, reconsidered. American Psychologist 1999, 54, (5), 317-326.
- Garcia-Cervantes, L.; Rodriguez-Romo, G.; Esteban-Cornejo, I.; Cabanas-Sanchez, V.; Delgado-Alfonso, A.; Castro-Pinero, J.; Veiga, O. L., Perceived environment in relation to objective and self-reported physical activity in Spanish youth. The UP&DOWN study. J Sports Sci 2016, 34, (15), 1423-9.
- Ortega, F. B.; Artero, E. G.; Ruiz, J. R.; Vicente-Rodriguez, G.; Bergman, P.; Hagstromer, M.; Ottevaere, C.; Nagy, E.; Konsta, O.; Rey-Lopez, J. P.; Polito, A.; Dietrich, S.; Plada, M.; Beghin, L.; Manios, Y.; Sjostrom, M.; Castillo, M. J., Reliability of health-related physical fitness tests in European adolescents. The HELENA Study. Int J Obes (Lond) 2008, 32 Suppl 5, S49-57.
- Mechanic, D.; Hansell, S., Adolescent competence, psychological well-being, and self-assessed physical health. J Health Soc Behav 1987, 28, (4), 364-74.
- Zullig, K. J.; Valois, R. F.; Huebner, E. S.; Drane, J. W., Adolescent health-related quality of life and perceived satisfaction with life. Qual Life Res 2005, 14, (6), 1573-84.
- Heard, H.; Gorman, B.; Kapinus, C., Family structure and selfrated health in adolescence and young adulthood. Population Research and Policy Review. 2008, 27(6), 773-797.
NOTE: Please, see tables and figures in the attached file.
Table Supplementary 1. Multicollinearity analysis.
|
Baseline |
||||||
|
|
Children (n=687) |
Adolescents (n=691) |
||||
|
VIF |
VIF |
|||||
|
Model 1 |
|
|
|
|
|
|
|
Cardiorespiratory fitness |
|
1.67 |
|
|
1.80 |
|
|
UIMS/WG |
|
1.06 |
|
|
1.32 |
|
|
Lower body explosive strength |
|
1.65 |
|
|
1.99 |
|
|
Motor fitness |
|
1.51 |
|
|
1.76 |
|
|
Global physical fitness |
|
1.13 |
|
|
1.11 |
|
|
Model 2 |
|
|||||
|
Cardiorespiratory fitness |
|
2.00 |
|
|
2.47 |
|
|
UIMS/WG |
|
1.74 |
|
|
1.94 |
|
|
Lower body explosive strength |
|
2.68 |
|
|
3.95 |
|
|
Motor fitness |
|
2.73 |
|
|
3.22 |
|
|
2-years follow-up |
||||||
|
|
Children (n=613) |
Adolescents (n=496) |
||||
|
VIF |
VIF |
|||||
|
Model 3 |
|
|
|
|
|
|
|
Cardiorespiratory fitness |
|
1.60 |
|
|
1.82 |
|
|
UIMS /WG |
|
1.09 |
|
|
1.33 |
|
|
Lower body explosive strength |
|
1.61 |
|
|
1.92 |
|
|
Motor fitness |
|
1.50 |
|
|
1.71 |
|
|
Global physical fitness |
|
1.12 |
|
|
1.10 |
|
|
Model 4 |
|
|
|
|
|
|
|
Cardiorespiratory fitness |
|
2.00 |
|
|
2.63 |
|
|
UIMS /WG |
|
1.74 |
|
|
1.97 |
|
|
Lower body explosive strength |
|
2.69 |
|
|
4.00 |
|
|
Motor fitness |
|
2.75 |
|
|
3.25 |
|
VIF: Variance inflation factor; UIMS/WG: Upper isometric muscular strength divided by body weight.
Table Supplementary 2. Likelihood ratio test for cumulative link models
|
Baseline |
||||||||||
|
|
Children (n=687) |
Adolescents (n=691) |
||||||||
|
ΔAIC |
ΔLogLik |
P |
ΔAIC |
ΔLogLik |
P |
|||||
|
Model 1 |
|
|
|
|
|
|
||||
|
Cardiorespiratory fitness |
-7.243 |
4.621 |
0.002 |
-21.635 |
11.817 |
<0.001 |
||||
|
UIMS /WG |
-6.698 |
4.349 |
0.003 |
-1.369 |
1.684 |
0.066 |
||||
|
Lower body explosive strength |
1.801 |
0.099 |
0.656 |
1.070 |
0.464 |
0.335 |
||||
|
Motor fitness |
0.921 |
0.539 |
0.299 |
-1.909 |
1.955 |
0.048 |
||||
|
Muscular strength score |
-13.053 |
7.527 |
<0.001 |
-0.828 |
1.414 |
0.093 |
||||
|
Global physical fitness |
-12.797 |
7.399 |
<0.001 |
-10.054 |
6.027 |
<0.001 |
||||
|
Model 2 |
|
|||||||||
|
Cardiorespiratory fitness |
-17.460 |
9.730 |
<0.001 |
-37.966 |
19.983 |
<0.001 |
||||
|
UIMS /WG |
0.054 |
0.973 |
0.163 |
1.592 |
0.204 |
0.523 |
||||
|
Lower body explosive strength |
1.874 |
0.063 |
0.723 |
1.539 |
0.230 |
0.497 |
||||
|
Motor fitness |
1.993 |
0.003 |
0.937 |
1.059 |
0.471 |
0.332 |
||||
|
2-years follow-up |
||||||||||
|
|
Children (n=613) |
Adolescents (n=496) |
||||||||
|
ΔAIC |
ΔLogLik |
P |
ΔAIC |
ΔLogLik |
P |
|||||
|
Model 3 |
|
|
|
|
|
|
||||
|
Cardiorespiratory fitness |
-8.860 |
5.430 |
<0.001 |
-1.519 |
1.760 |
0.060 |
||||
|
UIMS/WG |
1.295 |
0.352 |
0.401 |
1.292 |
0.354 |
0.400 |
||||
|
Lower body explosive strength |
0.948 |
0.525 |
0.305 |
0.072 |
0.964 |
0.165 |
||||
|
Motor fitness |
1.919 |
0.040 |
0.776 |
1.647 |
0.177 |
0.552 |
||||
|
Muscular strength score |
-5.200 |
3.598 |
0.007 |
0.051 |
0.974 |
0.163 |
||||
|
Global physical fitness |
-6.007 |
4.003 |
0.005 |
-0.624 |
1.312 |
0.105 |
||||
|
Model 4 |
|
|
|
|
|
|
||||
|
Cardiorespiratory fitness |
-17.280 |
9.640 |
<0.001 |
-2.602 |
2.301 |
0.031 |
||||
|
UIMS/WG |
1.146 |
0.427 |
0.355 |
0.982 |
0.509 |
0.313 |
||||
|
Lower body explosive strength |
0.005 |
0.997 |
0.158 |
-2.582 |
2.291 |
0.032 |
||||
|
Motor fitness |
1.350 |
0.325 |
0.420 |
2.000 |
0.000 |
0.999 |
||||
ΔAIC: Increment in Akaike Information Criterion; ΔLogLik: Increment in log-likelihood; UIMS/WG: Upper isometric muscular strength divided by body weight.
NOTE: Please, see tables and figures in the attached file.
Figure 1. Residual analysis for model 1.
Figure 2. Residual analysis for model 2 in children (Youngers).
Figure 3. Residual analysis for model 2 in adolescents (Olders).
Figure 4. Residual analysis for model 3.
Figure 5. Residual analysis for model 4 in children (Youngers).
Figure 6. Residual analysis for model 4 in adolescents (Olders).

Reviewer 2 Report
REVIEWER COMMENTS 736858
Cardiorespiratory fitness is a predictor of present and future self-rated health in youth
The manuscript aimed to examine the independent and combined associations of components of physical fitness with self-rated health at baseline (cross-sectional) and 2 years later (longitudinal) in a sample of 1378 Spanish children and adolescents (aged 8 to 17.9 years). Tests for physical fitness included 20-m shuttle run (cardiorespiratory fitness), handgrip strength divided by body weight, standing long jump (both for muscular fitness) and 4x10m shuttle run test (motor fitness). Also, a gender- and age- specific z-score was calculated as a global physical fitness score. Self-rated health was measured using a single, validated question. The study addressed a relevant research topic and it meets the scope of the journal. However, methodological aspects were not suitable in the study or were not reported in the text; thus, they imply issues on internal and external validity of the study. Specifically, information about the sampling and the procedures of estimating predictor variables (i.e., muscular fitness and global fitness scores) needs attention. Please see the specific comments.
Title:
- Considering reporting guidelines (i.e. STROBE), please clarify the study design in the title. Also, the title highlighted the main result of the study, but without relevant information about it (i.e., only for longitudinal data). I suggest using a title that describes all content of the study, or adjust the title in order to show the results precisely.
Abstract
- Please add information on the main statistical procedures that were used to test the hypothesis.
- Please add information on the statistical parameter (i.e., odds ratio or mean difference, as well their 95% confidence interval) for the main result of the study.
Introduction:
- Lines 52-42 and 62-64: The content of both sentences is similar. Please adjust it.
Method:
- Lines 76-77: Please clarify which type of bias (reporting bias?).
- Even information on the UP&DOWN study has been published elsewhere, data on selection procedures are missing. I suggest adding information on a few aspects of the population and sample: which were procedures adopted to invite the sample of the study? Which were the reasons for dropping out of the study? Was there any kind of attrition bias (for example, older adolescents vs. younger adolescents trend to be dropouts in the follow-up)? This information is relevant to show internal and external validity of the study sample.
- Lines 99-101: Please add information on the parameters of reliability and validity of the question for the same population – references 4, 29 and 30 did not support this information.
- LINES 113-124: Muscular strength was assessed based on maximum handgrip strength (upper body muscular strength) and the standing long jump (lower body explosive strength) tests. However, the tests were analyzed separately, while they seem to represent the same predictor of physical fitness (muscular fitness). Why were the tests analyzed separately? I believe they could be combined as a z-score for muscular fitness and tested in a singular score. I suggest testing whether a combined muscular fitness score is a significant predictor of self-rated health.
- Lines 125-131: Please clarify the measurement unit of motor fitness (seconds?).
- Lines 145-155: Was the global fitness score included in the statistical analyses with other physical fitness variables (for example, in model 2 and 4)? If so, was collinearity tested in order to avoid multicollinearity between predictor variables?
Results:
- Figures 1 and 2: please explain the information (letters “a” and “b”, p-value and the statistical test for it) in the footnotes of the illustration.
Discussion
- Lines 246-247: “Probably results of positive association between global physical fitness and SRH might be mediated by cardiorespiratory fitness.” Why did the authors not test this hypothesis?
- Please adjust the discussion and the conclusion according to the further results if necessary (combined muscular fitness score).
References:
- The manuscript included 51 references. Most of them were cited only once or were not included in the discussion of the results. I think the authors may reduce the number of references, focusing on the primary studies of the background and theoretical support of the scope/results.
Author Response
ANSWER TO THE REVIEWERS’ COMMENTS
REVIEWER - 2-
Comment
The manuscript aimed to examine the independent and combined associations of components of physical fitness with self-rated health at baseline (cross-sectional) and 2 years later (longitudinal) in a sample of 1378 Spanish children and adolescents (aged 8 to 17.9 years). Tests for physical fitness included 20-m shuttle run (cardiorespiratory fitness), handgrip strength divided by body weight, standing long jump (both for muscular fitness) and 4x10m shuttle run test (motor fitness). Also, a gender- and age- specific z-score was calculated as a global physical fitness score. Self-rated health was measured using a single, validated question. The study addressed a relevant research topic and it meets the scope of the journal. However, methodological aspects were not suitable in the study or were not reported in the text; thus, they imply issues on internal and external validity of the study. Specifically, information about the sampling and the procedures of estimating predictor variables (i.e., muscular fitness and global fitness scores) needs attention. Please see the specific comments.
Answer
Please, find a revised version of our manuscript entitled “Physical fitness and self-rated health in youth: cross-sectional and longitudinal study” (before: “Cardiorespiratory fitness is a predictor of present and future self-rated health in youth”). We would like to thank the associate Editor and the Reviewers for their time and the assessment of our manuscript and for providing us with this opportunity to improve the quality of our paper based on their constructive feedback.
An itemized point-by-point response to the reviewers’ comments is presented below. We have considered all of the Reviewers’ suggestions, and have either incorporated them into the revised manuscript or offered our rationale for not doing so. Changes to the original manuscript are highlighted directly in the text using a yellow font.
The authors are really delighted for having the opportunity of publishing with your journal.
Title:
Comment
Considering reporting guidelines (i.e. STROBE), please clarify the study design in the title. Also, the title highlighted the main result of the study, but without relevant information about it (i.e., only for longitudinal data). I suggest using a title that describes all content of the study, or adjust the title in order to show the results precisely.
Answer
Thank you very much to the reviewer for his constructive feedback. Tittle has been modified in the text as: “Physical fitness and self-rated health in youth: cross-sectional and longitudinal study”
Abstract
Comment
Please add information on the main statistical procedures that were used to test the hypothesis.
Please add information on the statistical parameter (i.e., odds ratio or mean difference, as well their 95% confidence interval) for the main result of the study.
Answer
We want to thank the reviewer for this interesting suggestion. We understand that including specific details about the main statistical procedures would be helpful. We originally had written an abstract that included these suggested details (odds ratio of the main results as well their 95% confidence interval) but were restricted by the required word limit of 200 words of the Journal. According reviewer´s suggestion abstract has been modified (See abstract section, page 1, lines 23-39).
We apologize to Editor because abstract is now 347 words we hope this will not be a problem.
ntroduction:
Comment
Lines 52-42 and 62-64: The content of both sentences is similar. Please adjust it.
Answer
Thank you for the suggestion. Authors apologize for the misunderstanding. We wanted to seek there were previous cross-sectional studies but they didn´t investigate the independent association of components of physical fitness with SRH in youth (line 54-56, before). And there are no longitudinal studies investigating the independent or combined associations of components of physical fitness with SRH in youth. (Lines 62-64, before). According reviewer´s suggestion text has been modified. (See introduction-section, page 2, lines 71-74).
Method:
Comment
Lines 76-77: Please clarify which type of bias (reporting bias?).
Even information on the UP&DOWN study has been published elsewhere, data on selection procedures are missing. I suggest adding information on a few aspects of the population and sample: which were procedures adopted to invite the sample of the study? Which were the reasons for dropping out of the study? Was there any kind of attrition bias (for example, older adolescents vs. younger adolescents trend to be dropouts in the follow-up)? This information is relevant to show internal and external validity of the study sample.
Answer
The authors acknowledge the Reviewer’s suggestions. We do apologize for the apparently biased description of the methodology.
For the feasibility of our study and for the follow-up study, we followed a strict standardization of the fieldwork. Considering the suggestion of the reviewer to improve the clarity of the methods section. We have provided more information in the manuscript about selection procedures, the number of eligible schools, how many schools agreed to participate, the number of students agreed to participate in the UP&DOWN study and losses due to adherence to the study protocol. (See Study design, settings and participants section, page 3, lines 86-93)
On the other hand, although there is always bias associated with self-report measures, we tried to minimize this through: 1) The use of standardized procedures and validated questionnaires used in previous studies in children [1-3] and adolescents [3-5]; 2) In children, the scientist in charge of this research tool, helped them one by one to clarify self-reported questions and rates in order to minimize possible bias because immaturity, making our estimations conservative [6] and 3) In addition, for the current study, participants aged <8 years did not complete self-reported measures about SRH, because there are not validated questionnaires for this range of age. Thus, they were not taking into account either in other papers of UP&DOWN study [7] were self-reported measures were required and not validated questionnaires were available for this range age. (See measurements: self-rated health (SRH)- section, page 4, line 128; Measurements (SRH)- section, page 4, lines 128-130; and Study design, settings and participants - section, page 3, lines 95-96)
Reasons for dropping out in our study (19,5%) were for those participants who didn´t complete data at baseline and follow-up on body mass index (BMI), pubertal status, SRH, cardiorespiratory fitness, muscular strength and motor fitness. We understand there wasn´t any kind of attrition bias which should affect the study taking into account it was a 3 year longitudinal study where 2.225 children and adolescents where tested. The research team worked hard for getting the dropping were as minor as possible doing extra ordinal sessions when some of participants didn´t attend the ordinal session by absence to the school.
Comment
Lines 99-101: Please add information on the parameters of reliability and validity of the question for the same population – references 4, 29 and 30 did not support this information.
Answer
Considering the reviewer´s suggestion the manuscript has been modified including references of standardized procedures and validated questionnaires used in previous studies in children [1-3] and adolescents [3-5] for measuring SRH. (See measurements: self rated health (SRH)-section, page 4, line 128)
Comment
LINES 113-124: Muscular strength was assessed based on maximum handgrip strength (upper body muscular strength) and the standing long jump (lower body explosive strength) tests. However, the tests were analyzed separately, while they seem to represent the same predictor of physical fitness (muscular fitness). Why were the tests analyzed separately? I believe they could be combined as a z-score for muscular fitness and tested in a singular score. I suggest testing whether a combined muscular fitness score is a significant predictor of self-rated health.
Answer
Thank you very much for your comment. A muscular strength score was calculated and analyzed as the reviewer suggest. However, based on the results obtained we believe it is important to show result of handgrip and standing long jump separately as the handgrip seems to be a significant predictor of SRH on baseline models while the standing long jump does not. Anyway, following reviewer´s comment a table with the muscular strength score has been included at the end of this document (see table Supplementary 1), and extra information has been added to table 2 and 3 in the manuscript. (See tables section, pages 8-11)
Comment
Lines 125-131: Please clarify the measurement unit of motor fitness (seconds?).
Answer
Authors apologize for the misunderstanding. Motor fitness was recorded in seconds. The information has been added in the manuscript. (See measurements: Physical fitness- section, page 4, line 159)
Comment
Lines 145-155: Was the global fitness score included in the statistical analyses with other physical fitness variables (for example, in model 2 and 4)? If so, was collinearity tested in order to avoid multicollinearity between predictor variables?
Answer
Thank you for the suggestion. Pubertal status, age and BMI were the only covariables when the global fitness score was included in the model so we did not to test Multicollinearity. Multicollinearity was analysed for model 2 and 4 due to the inclusion of all physical fitness variables in the model, but none of VIF calculated for independent variables was higher than 10 (see below table Supplementary 2). A sentence about collinearity analysis has been included in the statistical analysis section. (See Statistical analysis- section, page 5, lines 203-205)
Results:
Comment
Figures 1 and 2: please explain the information (letters “a” and “b”, p-value and the statistical test for it) in the footnotes of the illustration.
Answer:
A text has been added in the footnotes of the illustration clarifying the meaning letters “a” and “b”, p-value and the statistical test for it. (See figures section- page 12)
Discussion
Comment
Lines 287-288: “Probably results of positive association between global physical fitness and SRH might be mediated by cardiorespiratory fitness.” Why did the authors not test this hypothesis?
Answer
We want to thank the reviewer for this interesting suggestion. We suggested in the discussion section that the positive association between global physical fitness and SRH might be mediated by cardiorespiratory fitness. This hypothesis was based on the weight that cardiorespiratory fitness shows on the final results of our study, being the only one component of physical fitness, which was independently and positively associated with SRH in children (8-11.9 years age) and adolescents (12-17.9 years age) at baseline and 2-years follow-up.
For sure, we will take into account this interesting suggestion if we have the opportunity in our next research. Currently, our team research is in progress to develop U&DOWN study phase II where participants in UP&DOWN-I- will be measured 6 years later. We hope to get more information about how SRH and physical fitness modified from childhood to adolescence.
Comment
Please adjust the discussion and the conclusion according to the further results if necessary (combined muscular fitness score).
Answer
Suggested analysis by the reviewers 1 and 2 have been done and results have not changed. Thus, adjust on discussion and conclusion are not applicable. Thank you very much for the reminder.
References:
Comment
The manuscript included 51 references. Most of them were cited only once or were not included in the discussion of the results. I think the authors may reduce the number of references, focusing on the primary studies of the background and theoretical support of the scope/results.
Answer
We tried to follow reviewer´s comment. Accordingly, the introduction section (background) have been checked and several references have been deleted. Finally, the manuscript includes a total of 48 references. Authors apologize because on the other hand, some reviewers suggest to introduce some new references.
References
- Morales, P. F.; Sanchez-Lopez, M.; Moya-Martinez, P.; Garcia-Prieto, J. C.; Martinez-Andres, M.; Garcia, N. L.; Martinez-Vizcaino, V., Health-related quality of life, obesity, and fitness in schoolchildren: the Cuenca study. Qual Life Res 2013, 22, (7), 1515-23.
- Andersen, J. R.; Natvig, G. K.; Aadland, E.; Moe, V. F.; Kolotkin, R. L.; Anderssen, S. A.; Resaland, G. K., Associations between health-related quality of life, cardiorespiratory fitness, muscle strength, physical activity and waist circumference in 10-year-old children: the ASK study. Qual Life Res 2017, 26, (12), 3421-3428.
- Lämmle, L.; Woll, A.; Mensik, G.; Bös, K., Distal and proximal factors of health behaviors and their associations with health in children and adolescents. International Journal of envirommental research and public heatlh 2013,10, (7), 2944-78.
- Kantomaa, M. T.; Tammelin, T.; Ebeling, H.; Stamatakis, E.; Taanila, A., High levels of physical activity and cardiorespiratory fitness are associated with good self-rated health in adolescents. J Phys Act Health 2015, 12, (2), 266-72.
- Mota, J.; Santos, R. M.; Silva, P.; Aires, L.; Martins, C.; Vale, S., Associations between self-rated health with cardiorespiratory fitness and obesity status among adolescent girls. J Phys Act Health 2012, 9, (3), 378-81.
- Sanchez-Oliva, D.; Leech, R. M.; Grao-Cruces, A.; Esteban-Cornejo, I.; Padilla-Moledo, C.; Veiga, O. L.; Cabanas-Sanchez, V.; Castro-Pinero, J., Does modality matter? A latent profile and transition analysis of sedentary behaviours among school-aged youth: The UP&DOWN study. J Sports Sci 2020, 1-8.
- Castro-Pinero, J.; Carbonell-Baeza, A.; Martinez-Gomez, D.; Gomez-Martinez, S.; Cabanas-Sanchez, V.; Santiago, C.; Veses, A. M.; Bandres, F.; Gonzalez-Galo, A.; Gomez-Gallego, F.; Veiga, O. L.; Ruiz, J. R.; Marcos, A., Follow-up in healthy schoolchildren and in adolescents with Down syndrome: psycho-environmental and genetic determinants of physical activity and its impact on fitness, cardiovascular diseases, inflammatory biomarkers and mental health; the UP&DOWN study. BMC Public Health 2014, 14, 400.
NOTE: Please see tables in the attached file
Table Supplementary 1. Cumulative odd ratios and 95% confidence intervals for having high self-rated health according
to muscular fitness tests in children and adolescents.
|
Baseline |
||||||
|
|
Children (n=687) |
Adolescents (n=691) |
||||
|
OR |
95% CI |
p |
OR |
95% CI |
p |
|
|
UIMS/WG |
18.921 |
3.471 – 104.355 |
<0.001 |
5.707 |
1.122 – 29.205 |
0.036 |
|
Lower body explosive strength |
1.002 |
0.996 – 1.008 |
0.535 |
1.004 |
0.999 – 1.010 |
0.112 |
|
Muscular strength score |
1.216 |
1.101 – 1.343 |
<0.001 |
1.112 |
1.002 – 1.235 |
<0.05 |
|
2-years follow-up |
||||||
|
|
Children (n=613) |
Adolescents (n=496) |
||||
|
OR |
95% CI |
P |
OR |
95% CI |
P |
|
|
UIMS/WG |
2.154 |
0.344 – 13.519 |
0.412 |
0.981 |
0.135 – 7.164 |
0.985 |
|
Lower body explosive strength |
1.004 |
0.996 – 1.012 |
0.326 |
0.996 |
0.989 – 1.002 |
0.164 |
|
Muscular strength score |
1.110 |
1.105 – 1.313 |
0.052 |
1.100 |
0.988 – 1.226 |
0.083 |
OR: Cumulative odd ratios; CI: confidence intervals; UIMS/WG: upper body isometric muscular strength divided by body weight.
Baseline model: each model included one fitness test score separately. Covariates: gender, BMI and pubertal status.
2-years follow-up model: each model included one fitness test score at baseline separately. Covariates: gender, BMI and pubertal status at 2-year follow up and levels of SRH at baseline.
Muscular fitness score was calculated as the sum of UIMS/WG and lower body explosive strength z-scores
BMI was removed as covariate in the models that included UIMS/WG as independent variable.
Table Supplementary 2. Multicollinearity analysis.
|
Baseline |
||||||
|
|
Children (n=687) |
Adolescents (n=691) |
||||
|
VIF |
VIF |
|||||
|
Model 1 |
|
|
|
|
|
|
|
Cardiorespiratory fitness |
|
1.67 |
|
|
1.80 |
|
|
UIMS/WG |
|
1.06 |
|
|
1.32 |
|
|
Lower body explosive strength |
|
1.65 |
|
|
1.99 |
|
|
Motor fitness |
|
1.51 |
|
|
1.76 |
|
|
Global physical fitness |
|
1.13 |
|
|
1.11 |
|
|
Model 2 |
|
|||||
|
Cardiorespiratory fitness |
|
2.00 |
|
|
2.47 |
|
|
UIMS/WG |
|
1.74 |
|
|
1.94 |
|
|
Lower body explosive strength |
|
2.68 |
|
|
3.95 |
|
|
Motor fitness |
|
2.73 |
|
|
3.22 |
|
|
2-years follow-up |
||||||
|
|
Children (n=613) |
Adolescents (n=496) |
||||
|
VIF |
VIF |
|||||
|
Model 3 |
|
|
|
|
|
|
|
Cardiorespiratory fitness |
|
1.60 |
|
|
1.82 |
|
|
UIMS /WG |
|
1.09 |
|
|
1.33 |
|
|
Lower body explosive strength |
|
1.61 |
|
|
1.92 |
|
|
Motor fitness |
|
1.50 |
|
|
1.71 |
|
|
Global physical fitness |
|
1.12 |
|
|
1.10 |
|
|
Model 4 |
|
|
|
|
|
|
|
Cardiorespiratory fitness |
|
2.00 |
|
|
2.63 |
|
|
UIMS /WG |
|
1.74 |
|
|
1.97 |
|
|
Lower body explosive strength |
|
2.69 |
|
|
4.00 |
|
|
Motor fitness |
|
2.75 |
|
|
3.25 |
|
VIF: Variance inflation factor; UIMS/WG: upper body isometric muscular strength divided by body weight.

Round 2
Reviewer 1 Report
Dear authors,
Thank you for taking the time to revise the manuscript, it is much improved and I hope you are satisfied as well. This is acceptable as it is, but you may wish to consider some minor, mostly stylistic suggestions below.
Four less laborious things for your consideration.
- For the title, please do not use "youth" but rather a bit longer "children and adolescents 8-17 year of age" or similar. I think this is a similar thing to "youngers" and "olders" Please check all.
- For the abstract, please place the OR values immediately after each variable, preferably not in a long list. A bit more reader friendly.
- L. 242 dot missing.
- Also for your response to my comments, you do not really need to respond to me, but please make sure all your response is in the manuscript. You could also highlight in the summary what you responded to me: Physical activity is important, but cardiorespiratory fitness especially.
I hope that you will continue on this important topic.
Reviewer 2 Report
The revised version of the manuscript addressed all the suggestions that I have made previously, improving the scientific quality of the final report. Congratulations on the relevant manuscript.